# Effect of Salinity Stress on Phenolic Compounds and Antioxidant Activity in Halophytes *Spergularia marina* (L.) Griseb. and *Glaux maritima* L. Cultured *In Vitro*

**DOI:** 10.3390/plants12091905

**Published:** 2023-05-07

**Authors:** Artem Pungin, Lidia Lartseva, Violetta Loskutnikova, Vladislav Shakhov, Elena Popova, Liubov Skrypnik, Olesya Krol

**Affiliations:** MedBio Cluster, Immanuel Kant Baltic Federal University, Universitetskaya Str. 2, 236040 Kaliningrad, Russia; lida.lartseva@mail.ru (L.L.); violett.loskutnikova@gmail.com (V.L.); vladlyshakhov@gmail.com (V.S.); elena_popova97@mail.ru (E.P.); okrol@kantiana.ru (O.K.)

**Keywords:** halophytes, salinity, secondary metabolites, flavonoids, biological active compounds, *in vitro* propagation

## Abstract

The study of halophytes as sources of phenolic compounds, as well as conditions that further enhance the accumulation of biologically active compounds in them, is of particular interest. In this paper, the effect of different salinity levels (25–500 mM in the form of NaCl) on the content of phenolic compounds and the antioxidant activity of two rare halophyte species *Spergularia marina* (L.) Griseb. and *Glaux maritima* L. cultured *in vitro* was investigated. A species-specific reaction of plants to salinization was established. In *G. maritima*, the maximum total content of phenolic compounds was observed at 50–100 mM, flavonoids 75–400 mM, and hydroxycinnamic acids 200–300 mM, as well as individual phenolics (protocatechuic acid, catechin, astragalin, hyperoside, rutin, isoquercitrin, and apigenin derivative) at 100–300 mM NaCl. For *S. marina*, on the contrary, there was a slight decrease in the content of phenolic compounds when NaCl was added to the nutrient medium compared to the control. The content of protocatechuic acid, rosmarinic acid, and apigenin derivative significantly decreased with increased salt stress. The change in antioxidant activity at different salinity levels was also species specific. The maximum values of different groups of phenolic compounds in *G. maritima* were observed at 50–300 mM NaCl. The cultivation of *S. marina* without the addition of NaCl and at 500 mM NaCl allowed the production of plants with the highest content of phenolic compounds. The obtained results can be further used in the development of protocols for the cultivation of these plants *in vitro* in order to induce the biosynthesis of phenolic compounds in them.

## 1. Introduction

Soil salinization is a global environmental problem. Soil salinization, both natural and secondary in irrigated agriculture, is one of the factors that strengthens the desertification process and threatens agriculture [1]. Soil salinization is a factor affecting plant nutrition and growth processes due to increased osmotic pressure in the soil [2]. Most plants do not tolerate the high concentration of salts in the soil and cannot grow on soils exposed to salt; such plants are called glycophytes [3]. The osmotic effect of salinity causes metabolic changes in plants similar to those associated with water scarcity [4]. In addition, ion toxicity and dietary imbalances resulting from salt stress reduce plant growth [5]. It is known that under conditions of salt stress in different vegetable crops, there is a decrease in biomass, leaf area, growth rate, and yield [6,7].

In turn, halophyte plants can survive and complete their life cycle at a salt concentration of more than 200 mM [8]. High salt concentrations, at which 99% of nonhalophytic plants die, contribute to the vegetative and reproductive development of halophytes [9]. Most known plant families contain halophytes (Amaranthaceae Juss., Asteraceae Bercht., J.Presl, Nitrariaceae Lindl., Caryophyllaceae Juss., and others), but they make up only about 1% of all plant organisms on Earth [10]. 

Halophytes have many different adaptation mechanisms for living in saline conditions. These include ion compartmentalization, selective transport and uptake of ions, osmotic adjustment through osmolytes accumulation, succulence, enzymatic and nonenzymatic antioxidant response, maintenance of redox and energetic status, salt inclusion/excretion, and genetic control [8,11]. The formation of reactive oxygen species (ROS) occurs under stress conditions. ROSs are very toxic and can cause oxidative damage to proteins, DNA, lipids, and cells in general in the absence of a defense mechanism in the plant [12]. To regulate ROS levels in plant cells, complex enzymatic and nonenzymatic mechanisms of antioxidant defense develop, which together help to control the redox state of cells under stress conditions [11]. In halophytes, the enzymatic antioxidant mechanism was found to be highly effective, which decreases the ROS level to a greater extent and supports the survival of plants under salt stress [11,13].

Secondary plant metabolites, as components of nonenzymatic antioxidant defense, play an important role in protecting plants under salt stress [14,15,16,17]. Saline stress is a factor that triggers signaling pathways of secondary metabolite synthesis to increase plant resistance to stress conditions [18], but this is crucially dependent on the salinity sensitivity of specific plant species [19]. The content of polyphenolic compounds in halophytes is higher than in glycophytes, and extracts from these plants are able to prevent the harmful effects of free radicals [20]. For example, for the halophyte *Cakile maritima* Scop., a significant increase in the accumulation of polyphenols and antioxidant activity of extracts with an increase in salinity was shown [21].

Two rare halophyte species, *Spergularia marina* (L.) Griseb. and *Glaux maritima* L., growing on the Baltic Sea coast and in lagoons, were chosen to study the effect of salt [22]. The annual *S. marina* is an obligate halophyte, growing on soils with variable but usually high salinity [23]. Perennial *G. maritima* prefers to grow in wet and salty places, which can also be periodically flooded [24]. These species have high biotechnological and therapeutic potential [25,26,27,28]. It was previously shown how the growth conditions of these species, in particular the level of soil salinity [22], affected the qualitative and quantitative composition of secondary metabolites. There is limited information on the effect of salt stress on *G. maritima* salinity tolerance under tissue-culture conditions and changes in the activity of oxidative enzymes as indicators of salinity-related processes [29]. In turn, no similar studies have been conducted previously for *S. marina*.

Due to the fact that these species are rare in the Baltic region [30,31], it is promising to develop biotechnological techniques for microclonal reproduction and cultivation *in vitro*, as well as obtaining cell and tissue cultures of these species. These approaches can be used both to preserve rare species and to study the ability to synthesize secondary metabolites in tissue culture, including when modeling various stress conditions, including salt stress. Based on this, the purpose of this study was to study the effect of salt stress on phenolic compounds and the antioxidant activity of halophytes *Spergularia marina* and *Glaux maritima* in the *in vitro* culture.

## 2. Results

### 2.1. Effect of Salinity on Biomass Production

A study of the effect of different concentrations of NaCl (25–500 mM) on the fresh weight of *G. maritima* and *S. marina* shoots grown *in vitro* showed a difference in the response of these plant species to salinization (Figure 1). 

The highest crude mass of shoots in *G. maritima* plants was (ANOVA, F = 62.1, df_1_ = 8, df_2_ = 27, *p* ≤ 0.001) observed in the absence of salinity (0 mM NaCl). At the same time, with increasing salinity, the crude weight of *G. maritima* shoots decreased and was minimal at a concentration of 500 mM NaCl (Figure 1a). The difference with the control at this NaCl concentration was significantly more than 10 times (0.166 ± 0.009 and 0.014 ± 0.002 g/plant, respectively). For *S. marina* plants, the highest shoot biomass (ANOVA, F = 8.4, df_1_ = 8, df_2_ = 27, *p* ≤ 0.001) was observed at a concentration of 75–300 mM NaCl (Figure 1b). At 500 mM NaCl, a decrease in fresh-shoot mass was observed. However, even at this concentration of NaCl, the crude biomass of *S. marina* shoots was not significantly different from the control.

A somewhat different dependence was revealed in the study of the dry weight of the shoots of *G. maritima* and *S. marina* (Figure 2). For both species studied, there was a slight difference in dry weight at low NaCl concentrations (up to 25–75 mM), compared to the controls. At higher salinity levels, a decrease in the dry weight of shoots was observed. At the maximum concentration of NaCl (500 mM), the dry weight of the shoots of *G. maritima* was four times lower (ANOVA, F = 26.2, df_1_ = 8, df_2_ = 27, *p* ≤ 0.001) and *S. marina* two times lower (ANOVA, F = 25.4, df_1_ = 8, df_2_ = 27, *p* ≤ 0.001) compared to the control. 

In general, it is worth noting that the *G. maritima* and *S. marina* plants under conditions of high salinity at concentrations of 300–500 mM NaCl had obvious signs of depression as a result of the stress reaction, which manifested themselves not only in a decrease in the raw and dry mass of the shoots but also had morphological manifestations, such as a decrease in the size of the shoots and a decrease in the size of the roots to their complete absence at 500 mM NaCl (Figure 3).

### 2.2. Effect of Salinity on the Content of Phenolic Compounds

The results showed that the total content of phenolic compounds (TPC) in *G. maritima* plants cultured for 40 days on media with different concentrations of NaCl was influenced by salt stress. The total phenolic-compounds content increased as the salinity of 0–75 mM NaCl increased. It can be seen that the highest content (8.9 ± 1.0 mg GAE g^−1^ DW) was observed at a concentration of 75 mM NaCl (ANOVA, F = 39.3, df_1_ = 8, df_2_ = 27, *p* ≤ 0.001) (Figure 4a). However, with the subsequent increase in salinity of 100–500 mM NaCl, the total content of phenolic compounds in *G. maritima* plants decreased to a minimum content (2.1 ± 0.3 mg GAE g^−1^ DW) at 500 mM NaCl.

In turn, the highest total content of phenolic compounds (4.9 ± 1.1 mg GAE g^−1^ DW) in *S. marina* plants (ANOVA, F = 8.5, df_1_ = 8, df_2_ = 27, *p* ≤ 0.001) was detected in the control (0 mM NaCl). Under salinity conditions, the content of phenolic compounds was significantly lower from 25–400 mM NaCl compared to the control and ranged from (2.6 ± 0.2 mg GAE g^−1^ DW) at a concentration of 200 mM NaCl to (3.9 ± 0.4 mg GAE g^−1^ DW) at 500 mM NaCl (Figure 4b).

Analysis of the total flavonoid content (TFC) in *G. maritima* plants showed the presence of the influence of salt stress. Thus, the total content of flavonoids in the control (0 mM NaCl) was minimal (1.3 ± 0.2 mg RE g^−1^ DW). With an increase in salinity to 100 mM NaCl, the flavonoid content significantly increased (ANOVA, F = 36.9, df_1_ = 8, df_2_ = 27, *p* ≤ 0.001) to the maximum (10.1 ± 1.1 mg RE g^−1^ DW) (Figure 5a). However, at a concentration of 500 mM NaCl, the flavonoid content was significantly reduced (5.1 ± 0.4 mg RE g^−1^ DW) but remained higher compared to the control.

The total flavonoid content in *S. marina* plants did not change significantly (ANOVA, F = 2.9, df_1_ = 8, df_2_ = 27, *p* ≤ 0.05) with increasing NaCl concentration (Figure 5b) and varied from the highest value in the control (1.9 ± 0.3 mg RE g^−1^ DW) to the lowest at a concentration of 100 and 200 mM NaCl (1.0 ± 0.3 and 1.1 ± 0.4 mg RE g^−1^ DW, respectively).

The effect of salinity on the total content of hydroxycinnamic acids (THA) in *G. maritima* plants is also shown (Figure 6a). With the increase in the NaCl concentration in the nutrient medium from 0 mM to 300 mM, the total content of hydroxycinnamic acids increased from 0.7 ± 0.3 mg CAE g^−1^ DW to the highest content of 3.2 ± 0.2 mg CAE g^−1^ DW (ANOVA, F = 25.7, df_1_ = 8, df_2_ = 27, *p* ≤ 0.001). As can be seen, a decrease in the content of hydroxycinnamic acids is observed at high concentrations of sodium chloride (400–500 mM). 

As a result of the analysis, it was not possible to establish significant differences (ANOVA, F = 2.0, df_1_ = 8, df_2_ = 27, *p* = 0.079) in the total content of hydroxycinnamic acids in *S. marina* plants at different salinity levels of 0–500 mM NaCl (Figure 6b).

### 2.3. The Effect of Salinity on the Antioxidant Activity

The effect of salinity and antioxidant activity of *G. maritima* extracts measured by DPPH (2,2-diphenyl-1-picrylhydrazyl), ABTS (2,2′-azino-bis(3-ethylbenzothiazoline-6-sulfonic acid)), and FRAP (ferric reducing/antioxidant power) assays was established (Figure 7a,c,e). Thus, the maximum antioxidant activity (ANOVA, F = 39.7, df_1_ = 8, df_2_ = 27, *p* ≤ 0.001) according to the DPPH assay was observed at a concentration of 100 and 500 mM NaCl (5.9 ± 0.4 and 5.8 ± 0.7 mg AscAc g^−1^ DW, respectively). The greatest restorative capacity of the extracts of *G. maritima*, according to the FRAP assay (ANOVA, F = 15.2, df_1_ = 8, df_2_ = 27, *p* ≤ 0.001), was established at a concentration of 300 mM NaCl (3.7 ± 0.6 mg AscAc g^−1^ DW). In turn, the determination of antioxidant activity by the ABTS method showed the highest value (ANOVA, F = 12.1, df_1_ = 8, df_2_ = 27, *p* ≤ 0.001) at a concentration of 75 mM NaCl (22.5 ± 1.8 mg AscAc g^−1^ DW).

Analysis of the antioxidant activity of *S. marina* extracts also showed a response to salinity (Figure 7b,d,f). The maximum antioxidant activity, according to the DPPH assay, (ANOVA, F = 19.8, df_1_ = 8, df_2_ = 27, *p* ≤ 0.001) was observed at a concentration of 500 mM NaCl (0.6 ± 0.2 mg AscAc g^−1^ DW), as well as the restorative capacity of the extracts according to the FRAP assay (1.0 ± 0.1 mg AscAc g^−1^ DW) (ANOVA, F = 31.9, df_1_ = 8, df_2_ = 27, *p* ≤ 0.001). Determination of antioxidant activity by ABTS showed a significant increase in activity (ANOVA, F = 18.2, df_1_ = 8, df_2_ = 27, *p* ≤ 0.001) at a concentration of 500 mM NaCl (5.5 ± 0.7 mg AscAc g^−1^ DW) while remaining below the activity value at a concentration of 0 mM NaCl (7.3 ± 1.4 mg AscAc g^−1^ DW).

### 2.4. Influence of Salinity on the Content of Individual Phenolic Compounds

Based on the analysis of *G. maritima* extracts by high-performance liquid chromatography (HPLC), it was found that the content of phenolic acids and flavonoids significantly depended on salt stress (Table 1). Among the phenolic acids in the samples, protocatechuic acid was found, as well as flavonoids such as catechin, astragalin, hyperoside, rutin, isoquercitrin, and apigenin derivative. The content of these substances in *G. maritima* samples significantly increased (ANOVA, *p* ≤ 0.001) to the highest values at a concentration of NaCl in the nutrient medium of 100–300 mM. Examples of chromatograms are shown in Figure A1 (Appendix A).

Based on the results of the analysis of *S. marina* extracts by HPLC, it was found that the content of protocatechuic acid (ANOVA, F = 437.1, df_1_ = 8, df_2_ = 27, *p* ≤ 0.001), rosmarinic acid (ANOVA, F = 500.9, df_1_ = 8, df_2_ = 27, *p* ≤ 0.001), and apigenin derivative (ANOVA, F = 82.5, df_1_ = 8, df_2_ = 27, *p* ≤ 0.001) significantly decreased with increasing salt stress (Table 2). A significant content of luteolin derivative (ANOVA, F = 1353.7, df_1_ = 8, df_2_ = 27, *p* ≤ 0.001) in *S. marina* samples was detected only in the control (0 mM NaCl) and at a concentration of 500 mM NaCl. Examples of the chromatograms are shown in Figure A2 (Appendix A).

### 2.5. Correlation between Antioxidant Activity, Phenolic Compounds Content, and Biomass Production

A high level of positive correlation was established between the dry mass of *G. maritima* shoots and the total content of phenolic compounds (*r* = 0.95; *p* ≤ 0.001), antioxidant activity determined using the ABTS method (*r* = 0.77; *p* ≤ 0.05), and the content of astragalin (*r* = 0.73; *p* ≤ 0.05) (Figure 8). A high positive correlation was found between the antioxidant activity of FRAP and the total content of flavonoids (*r* = 0.80; *p* ≤ 0.05) and hydroxycinnamic acids (*r* = 0.91; *p* ≤ 0.01), the content of quercetin derivative (*r* = 0.97; *p* ≤ 0.001), hyperoside (*r* = 0.92; *p* ≤ 0.001), and rutin (*r* = 0.97; *p* ≤ 0.001). A high negative relationship was established between the fresh weight of shoots and the total content of hydroxycinnamic acids (*r* = −0.73; *p* ≤ 0.05). However, there were no statistically significant correlations between the antioxidant activity determined by the DPPH method and the rest of the studied parameters.

The antioxidant activity of *S. marina* extracts (Figure 9), determined using the DPPH method, showed an average level of positive correlation with the total content of phenolic compounds (*r* = 0.74; *p* ≤ 0.05) and with the content of rosmarinic acid (*r* = 0.70; *p* ≤ 0.05), as well as a strong negative relationship with the crude mass of shoots (*r* = −0.90; *p* ≤ 0.01). A high positive correlation was found between the antioxidant activity of ABTS and the total content of phenolic compounds (*r* = 0.96; *p* ≤ 0.001) and with the content of rosmarinic acid (*r* = 0.98; *p* ≤ 0.001), as well as an average positive correlation with the total content of flavonoids (*r* = 0.69; *p* ≤ 0.05) and a strong negative relationship with the crude mass of shoots (*r* = −0.78; *p* ≤ 0.05). A strong negative association was established between the antioxidant activity of FRAP and the dry mass of shoots (*r* = −0.79; *p* ≤ 0.05), the content of protocatechuic acid (*r* = −0.74; *p* ≤ 0.05), and rosmarinic acid (*r* = −0.78; *p* ≤ 0.05). 

## 3. Discussion

In this study, two rare halophyte species in the Baltic region, *Spergularia marina* and *Glaux maritima* [30,31], were tested for *in vitro* salinity resistance, assessing the effect of salt stress on the accumulation of phenolic compounds and the antioxidant activity of halophyte extracts.

As growth parameters, to assess the resistance to salinity, the raw and dry mass of shoots was analyzed when growing plants at concentrations of 0–500 mM NaCl. When analyzing the fresh weight of the shoots, differences were found in the reaction of the plants *G. maritima* and *S. marina* to salinity. With increasing salinity, the crude weight of *G. maritima* shoots decreased, and at a concentration of 500 mM NaCl, the weight was less than 10 compared to the control. For *S. marina* plants, the highest crude-shoot biomass was observed at a concentration of 75–300 mM NaCl, and a decrease in weight was observed in the control and at 500 mM NaCl. In the dry-weight study of the shoots of *G. maritima* and *S. marina*, a slight difference in dry weight was noted at low NaCl concentrations (up to 25–75 mM) compared to the control. However, at higher salinity levels, there was a decrease in the dry weight of shoots in both species. Thus, at a concentration of 500 mM NaCl, the dry mass of *G. maritima* shoots was four times lower compared to the control, and the dry mass of *S. marina* shoots was two times lower. 

In a number of studies [21,32,33,34], when studying the effect of salinity on the growth characteristics of halophytes, the stimulating effect of low and moderate NaCl concentrations was noted, with a tendency to decrease in conditions of high salinity. For example, in three-week-old *Cakile maritima* Scop. plants, the seeds of which were collected from plants growing in arid bioclimatic conditions, when exposed to 0, 100, and 400 mM NaCl for 28 days in greenhouse conditions, an increase in shoot biomass and leaf expansion was noted at a concentration of 100 mM NaCl [21]. When studying the effects of salt stress (0–600 mM NaCl), the halophytes *Arthrocnemum macrostachyum* (Moric.) K. Koch and *Sarcocornia fruticosa* (L.) A.J. Scott showed the highest shoot and root weight in fresh and dry forms at 100 mM NaCl concentration; in turn, both species showed a significant decrease in these parameters at 600 mM NaCl [32]. In developing an *in vitro* micropropagation protocol for the medicinal halophyte *Limoniastrum monopetalum* (L.) Boiss, it was shown that adding NaCl at relatively low concentrations (2.5 or 5.0 g·L^−1^) to Murashige and Skoog nutrient media containing 0.5 mg·L^−1^ benzyladenine doubled shoot multiplication, but did not increase shoot elongation [35]. Relatively high tolerance to NaCl was found for *G. maritima* under tissue-culture conditions, and a stimulating effect was shown at a concentration of 100 mM NaCl, which manifested itself in the development of leaves, roots, and shoot elongation, and with an increase in salinity of 200–400 mM NaCl, significant growth inhibition occurred [31]. In our study, the obtained results did not show a pronounced stimulating effect of low and moderate concentrations of NaCl on the dry or crude biomass of the studied halophytes *G. maritima* and *S. marina*, while high concentrations led to a significant decrease in biomass and an inhibition of plant growth.

It is known that phenolic compounds are secondary metabolites that play an important role in protecting plants from oxidative stress, which can be caused by salt stress [16,17,36]. Phenolic compounds exhibit antioxidant activity, inactivating free-lipid radicals, and also prevent the decomposition of hydroperoxides into free radicals. The increased antioxidant promotes the detoxification of reactive oxygen species, which probably increases the resistance to salinity [32,37]. Our study showed the change in the content of various groups of phenolic compounds in *Glaux maritima* plants cultivated *in vitro* at different levels of salinity. The total content of phenolic compounds, the total content of flavonoids, as well as the content of astragalin and hyperoside, increased with an increase in salinity to 75–100 mM NaCl. The total content of hydroxycinnamic acids and the content of protocatechuic acid also increased with an increase in salinity to 200–300 mM NaCl. Salinization-induced changes in the content of various groups of phenolic compounds in *G. maritima* plants were closely correlated with a parallel change in the antioxidant activity of the extracts. Our earlier study of *G. maritima* growing in natural habitats showed an increase in the total content of phenols in shoots and hydroxycinnamic acids in plant roots in response to increased soil salinity and also showed a positive relationship between the antioxidant activity (DPPH) of shoot extracts with the total content of phenolic compounds (*r* = 0.700, *p* ≤ 0.05) and the total content of hydroxycinnamic acids (*r* = 0.691, *p* ≤ 0.05) [22].

It can be assumed that the synchronous increase in the antioxidant activity of extracts and the content of phenolic compounds, flavonoids, and hydroxycinnamic acids at moderate and high salinity indicates the importance of these compounds in the stress resistance of *G. maritima* plants. An increase in the content of polyphenols and an increase in the antioxidant activity of extracts with an increase in salinity was also noted in a number of plants, including halophytes. For example, it was shown that treatment with different concentrations of NaCl (10, 50, 100, and 200 mM) resulted in an increase in the amount of phenolic compounds and carotenoids in *Fagopyrum esculentum* Moench sprouts [38], as well as an increase in the antioxidant activity of the ethanol extracts of sprouts compared to the control (0 mM). In broccoli inflorescences, under the influence of moderate salt stress (40 mM NaCl), the content of polyphenolic compounds increased [19]. For the halophyte *Cakile maritima*, inhabiting arid climate conditions, a significant increase in polyphenol accumulation and antioxidant activity of extracts was shown with increasing salinity [21].

Thus, salt stress effectively increases the amount of phenolic compounds with antioxidant activity in *Glaux maritima* plants in contrast to *Spergularia marina*. In *S. marina* plants, the total content of phenolic compounds and flavonoids under saline conditions was lower compared to the control. The content of protocatechuic acid, rosmarinic acid, and apigenin derivative decreased with increasing salinity. However, an increase in rosmarinic acid and luteolin derivative was observed at high salinity (500 mM NaCl). Also, at high concentrations of NaCl, an increase in antioxidant activity (DPPH, FRAP, and ABTS) was observed. Our previous study showed an increase in total phenol content in *S. marina* roots and shoots and an increase in antioxidant activity (DPPH) of root extracts in response to increased soil salinity in natural habitats [22].

It is known that plants vary greatly in their qualitative and quantitative composition of phenolic compounds under saline conditions, and the content of phenolic compounds is determined both genetically and by the influence of environmental factors [20,32,39]. In a comparative study of the effect of salinity on the polyphenol content and antioxidant activity in the leaves of the halophyte *Cakile maritima* from two ecological populations [21], it was shown that the tolerance of plants grown in the dry bioclimatic zone (Jerba, Tunisia) to moderate processing of NaCl was accompanied by the enrichment of leaves with polyphenols, while the relative sensitivity to the salinity of plants grown in the wet bioclimatic zone (Tabarka, Tunisia) was accompanied by a decrease in the polyphenol content in the leaves. The authors [21] suggested that there was an intraspecific variability in the accumulation of polyphenols in response to salinity, and therefore explained the difference in the salt tolerance of these plants. The two samples may differ in their ability to compartmentalize salt in leaf cells or in their ability to maintain leaf–water balance [21]. When studying the salt resistance strategy in three Amaranthaceae halophytes from the same ecological habitat, the presence of two different strategies was noted [32]. One salt resistance strategy found in *Salicornia europaea* L. is driven by antioxidant enzyme activation and proline biosynthesis, and the second strategy is found in *Arthrocnemum macrostachyum* and *Sarcocornia fruticosa* is due to the rearrangement of the chlorophyll ratio and the biosynthesis of antioxidant compounds (carotenoids, phenols, and flavonoids), which require less energy than for the activation of antioxidant enzymes [32].

Thus, one can assume the presence of differential mechanisms of salt resistance in the studied species of the halophytes *Glaux maritima* and *Spergularia marina*, growing on the coast of the Baltic Sea and in lagoons. In this case, in *S. marina*, most likely, the mechanism of salt resistance is not associated with the biosynthesis of phenolic compounds and flavonoids. However, this issue merits further study.

## 4. Materials and Methods

### 4.1. Plant Material

Sterile plants were obtained from seeds collected in wild coastal populations near the Baltic Sea in the summer of 2020. The seeds of *Glaux maritima* and *Spergularia marina* were collected in a coastal meadow (54°37′ N, 19°53′ E). Preliminarily, the seeds were washed for 5 min with 5% (*v*/*v*) liquid detergent, then washed 5 times with sterile distilled water to remove any detergent residues. The sterilization of seeds was carried out in a laminar box (BMB-II-“Laminar-C”-1.5 (NEOTERIC), Class II Type A2 Biological Safety Cabinet, Lamsystems) using a solution of sodium hypochlorite (containing 1–2% active chlorine) with treatment for 15 min. The seeds were then washed 5 times with distilled water and placed to germinate in a Murashige and Skoog (MS) [40] medium-plated Petri dish supplemented with sucrose (3% *w/v*) and agar (0.7% *w/v*) without growth regulators. Seed germination was carried out in the dark at a temperature of 22 °C. *G. maritima* sterile seedlings were cultured in 100 mL Phytohealth tissue-culture containers (SPL Lifesciences, Naechon-Myeon, Republic of Korea) on MS medium supplemented with sucrose (3% *w/v*) and agar (0.7% *w/v*) without growth regulators and pH 5.6 to 5.8. In turn, *S. marina* sprouts were cultivated in the same containers on Gamborg (B5) [41] nutrient medium supplemented with sucrose (3% *w/v*) and agar (0.7% *w/v*) without growth regulators and pH from 5.6 to 5.8. To increase the amount of plant material, microcuttings of the *G. maritima* and *S. marina* cultures were carried out every 40 days. To do this, 15–20 mm long nodal and apical explants were planted on the appropriate nutrient medium mentioned above without growth regulators. Cultures were maintained in a growth room at 25 ± 2 °C using a 16 h photoperiod (white LED lamps; photosynthetic photon flux density of 160 μmol m^−2^ s^−1^). 

### 4.2. Experimental Setup

To conduct an experiment to assess the effect of salinity (NaCl) on the content of biologically active substances in the cultures of *G. maritima* and *S. marina*, nodal and apical explants with a length of 15–20 mm were used. *G. maritima* explants were placed on a growth regulator-free MS medium containing 3% sucrose (*w/v*) and 0.7% agar (*w/v*) and pH 5.6 to 5.8. *S. marina* explants were placed on a growth regulator-free B5 medium containing 3% sucrose (*w/v*) and 0.7% agar (*w/v*) and pH 5.6 to 5.8. NaCl at final concentrations of 0, 25, 50, 75, 100, 200, 300, 400, and 500 mM was added to the media for treatment.

Explants were cultured in 100 mL Phytohealth tissue-culture containers (SPL Lifesciences, Republic of Korea), 20 explants per vessel, and 4 vessels per NaCl concentration. Cultures were maintained in a growth room at 25 ± 2 °C using a 16 h photoperiod (white LED lamps; photosynthetic photon flux density of 160 μmol m^−2^ s^−1^). The duration of cultivation was 40 days.

### 4.3. Phytochemical Analysis

#### 4.3.1. Extract Preparation

After 40 days of cultivation, plant material (shoots with leaves) was collected. Fresh and dry biomasses were determined as the mean for 20 plants in one culture vessel. Plant material from each culture vessel for a given NaCl concentration was analyzed separately (n = 4). To determine the dry biomass, the plant material was dried in an oven at 60 °C for 24 h. The results of the determination of wet and dry biomass were expressed in g/plant.

Dry plant material was crushed to a particle size of less than 0.5 mm. The weighed amount of vegetable raw materials weighing 0.5 g was macerated at room temperature for 24 h with periodic shaking in 45 mL of a 70% ethanol solution. The extract was then filtered through a paper filter into a volumetric flask and the total volume was adjusted to 50 mL with a 70% ethanol solution.

#### 4.3.2. Determination of the Total Content of Certain Groups of Phenolic Compounds

The total content of phenolic compounds, flavonoids, and hydroxycinnamic acids was determined spectrometrically using a microplate reader (CLARIOstar, BMG Labtech, Ortenberg, Germany). All reactions were performed in a flat-bottomed 96-well microplate.

Total phenolic compounds (TPC) were determined using the Folin-Ciocalteu reagent [38]. One hundred microliters of the Folin-Ciocalteu reagent and 20 µL of the extract or standard were added to each well of the microplate. The mixture was stirred on an orbital shaker (MPS-1, BioSan, Riga, Latvia) and held for 4 min; then 75 μL of sodium carbonate (7.5%, *w/w*) was added. The mixture was incubated in the dark at room temperature for 2 h, then optical absorption was recorded at a wavelength of 765 nm. Gallic acid was used as the standard. The total content of phenolic compounds was determined according to the calibration graph (Appendix A, Table A1) and expressed in mg of gallic acid equivalents per gram of dry weight (mg GAE g^−1^ DW).

The total flavonoid content (TFC) was determined by complexation reaction with AlCl_3_ in the presence of sodium acetate [42] with modifications. The reaction mixture consisted of 50 µL of extract or standard, 10 µL of 10% AlCl_3_ solution, 10 µL of 1 M sodium acetate, and 150 µL of 96% ethanol. The mixture was stirred in an orbital shaker and incubated in the dark at room temperature for 40 min. Optical absorption was recorded at a wavelength of 415 nm. Rutin was used as the standard. The total flavonoid content was determined from the calibration graph (Appendix A, Table A1) and expressed in mg of rutin equivalents per gram of dry weight (mg RE g^−1^ DW).

The total content of hydroxycinnamic acids (THA) was determined using the Arno reagent [43] with modifications. To each well of the microplate were added 20 μL of extract or standard, 40 μL of 0.5 M HCl, 40 μL of Arno reagent (a mixture of 10% NaNO_2_ and 10% NaMoO_4_ in a ratio of 1:1), 40 μL of sodium hydroxide (8.5%, *w/w*), and 60 μL of distilled water. Chlorogenic acid was used as the standard; in turn, optical absorption was recorded at a wavelength of 525 nm. The total content of hydroxycinnamic acids was determined according to the calibration graph (Appendix A, Table A1), and for the plant *G. maritima* expressed in mg equivalents of chlorogenic acid per gram of dry weight (mg CAE g^−1^ DW), in turn for the plant *S. marina* expressed in mg equivalents of rosmarinic acid per gram of dry weight (mg RA g^−1^ DW).

#### 4.3.3. Determination of Antioxidant Activity

The antioxidant activity of *G. maritima* and *S. marina* extracts was determined by the ability to capture the radicals 2,2-diphenyl-1-picrylhydrazyl (DPPH) and 2,2′-azino-bis (3-ethylbenzthiazolino-6-sulfonic acid (ABTS) and by the reducing ability when reacting with Fe(III) -2,4,6-tripyridyl-s-triazine complex (FRAP) according to [44]. Antioxidant activity by DPPH, ABTS, and FRAP was determined spectrometrically using a microplate reader (CLARIOstar, BMG Labtech, Germany). All reactions were performed in a flat-bottomed 96-well microplate. As a standard, solutions of ascorbic acid of known concentration were used (Appendix A, Table A1). The assay results are expressed in mg ascorbic acid equivalents per gram of dry weight (mg AscAc g^−1^ DW).

To determine the antioxidant activity by DPPH, 300 μL of 0.1 mM DPPH solution and 20 μL of the extract or standard solution were added to each well of the microplate. The mixture was incubated in the dark at room temperature for 60 min. A reduction in optical absorption compared to the control was recorded at 515 nm.

When determining the antioxidant activity of ABTS, a solution of the ABTS radical was preliminarily prepared. The ABTS radical was prepared by mixing aliquots of 7.0 mM ABTS solution and 2.45 mM of potassium persulfate solution. The resulting solution was kept for 16 h in the dark at room temperature. To carry out the reaction, 300 μL of the prepared solution of the cation radical ABTS and 20 μL of the extract or standard were added to each well of the microplate. The resulting mixture was incubated for 15 min at 37 °C in the dark and optical absorption was measured at a wavelength of 734 nm.

Freshly prepared FRAP reagent was used to determine the restorative capacity of the extracts. The FRAP reagent was obtained by mixing 10 parts of 0.3 M acetate buffer (pH 3.6), one part of 10 mM solution of 2,4,6-tripyridyl-s-triazine in 40 mM HCl, and one part of 20 mM aqueous iron chloride solution of FeCl_3_·6H_2_O. To carry out the reaction, 300 μL of FRAP reagent and 20 mL of the test extract or standard solution were added to each well of the microplate. The resulting mixture was incubated for 10 min at 37 °C in the dark; optical absorption was measured at a wavelength of 593 nm.

#### 4.3.4. The Determination of Individual Phenolic Compounds

Individual phenolic compounds were determined by high-performance liquid chromatography with diode-array detection (HPLC-DAD) [44]. Before analysis, the extracts, prepared as described above, were filtered and concentrated on a rotary evaporator. The resulting dry matter was then dissolved in a 10% methanol solution. The new extract was centrifuged (4500× *g*) for 15 min, and the supernatant was filtered through a syringe filter (0.22 µm). The separation was performed on a Shimadzu LC-20 Prominence chromatograph with a Shimadzu SPD20MA diode array detector and a Phenomenex Luna column (C18 250 × 4.6 mm, 5 µm). The mobile phase consisted of a mixture of water/trifluoroacetic acid solvents 99.9/0.1 (solvent A) and acetonitrile (solvent B). Gradient mode, flow rate 1.0 mL min^−1^, column temperature 40 °C, and sample volume 10 µL were used to separate the substances. Detection was carried out in the wavelength range of 180–900 nm. The compounds were identified in the LabSolutions program by comparing the retention times of the peaks and UV spectra obtained in the chromatograms with the corresponding parameters of the chromatographically pure standards of the samples. Quantitative analysis of the studied flavonoids was carried out according to calibration plots built in the concentration range of 10–100 µg mL^−1^. The following Sigma-Aldrich standards (Sigma-Aldrich Rus, Moscow, Russia) were used: catechin, hyperoside, quercetin 3-*O*-rutinoside, quercetin 3-β-d-glucoside, kaempferol 3-*O*-glucoside, 3,4-dihydroxybenzoic acid, rosmarinic acid, apigenin 7-*O*-glucoside, and luteolin 7-*O*-glucoside.

### 4.4. Statistical Analysis

IBM SPSS Statistics 23 was used for the statistical processing of the data obtained. A one-way analysis of variance (ANOVA), followed by Tukey’s HSD test, was used to test the statistical hypotheses and estimate the validity of differences (*p* ≤ 0.05). Statistical processing of data was carried out by taking into account biological replications (*n* = 4). Statistical results are presented as mean ± standard deviation. Pearson’s correlation coefficient (*p* ≤ 0.05) was used to evaluate the correlation of quantitative traits.

## 5. Conclusions

We studied the effect of various salinity levels (25–500 mM in the form of NaCl) on the content of phenolic compounds and the antioxidant activity of the extracts of two rare species of halophytes, *S. marina* and *G. maritima*, cultivated *in vitro*. Our data show the presence of a species-specific response of plants to salinity, both in terms of the accumulation of phenolic compounds and changes in the antioxidant activity of extracts. In *G. maritima*, the maximum total content of phenolic compounds was observed at 50–100 mM, flavonoids at 75–400 mM, and hydroxycinnamic acids at 200–300 mM. For *S. marina*, on the contrary, there was a slight decrease in the content of phenolic compounds during salinization compared to the control. The maximum values of the antioxidant activity of *G. maritima* extracts were observed at medium NaCl concentrations (50–300 mM), while for *S. marina* extracts, the maximum values were obtained either in the control or at a maximum NaCl concentration of 500 mM.

In general, the increase in phenolic compounds and antioxidant activity in *Glaux maritima* under moderate salt stress confirms that halophytes, as plants resistant to the salt stress of plants can potentially be used in biotechnology for the biosynthesis of valuable secondary metabolites useful for various industries. Thus, at 75–100 mM NaCl, an increase in the yield of total phenolic compounds by about 40%, flavonoids by 675%, and hydroxycinnamic acids at 200–300 mM NaCl by 350% (compared with control conditions) was achieved. 

## Figures and Tables

**Figure 1 plants-12-01905-f001:**
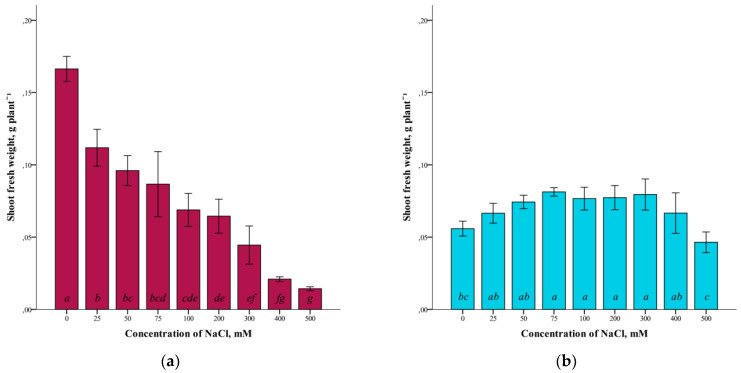
Shoot fresh weights *G. maritima* (**а**) and *S. marina* (**b**) under different NaCl concentrations. Different letters indicate significant differences (ANOVA, Tukey-HSD, *p* ≤ 0.05, *n* = 4).

**Figure 2 plants-12-01905-f002:**
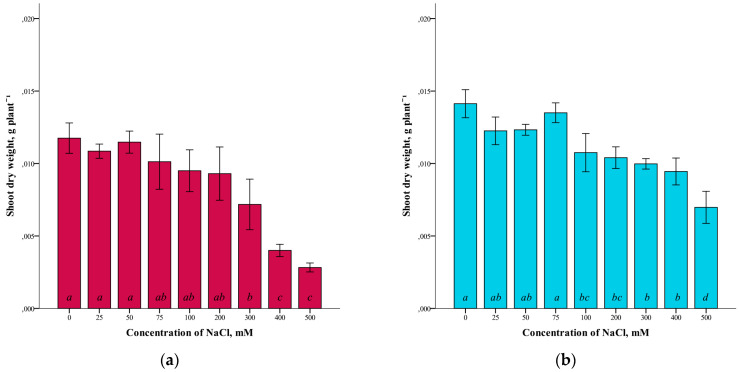
Shoot dry weights *G. maritima* (**а**) and *S. marina* (**b**) under different NaCl concentrations. Different letters indicate significant differences (ANOVA, Tukey-HSD, *p* ≤ 0.05, *n* = 4).

**Figure 3 plants-12-01905-f003:**
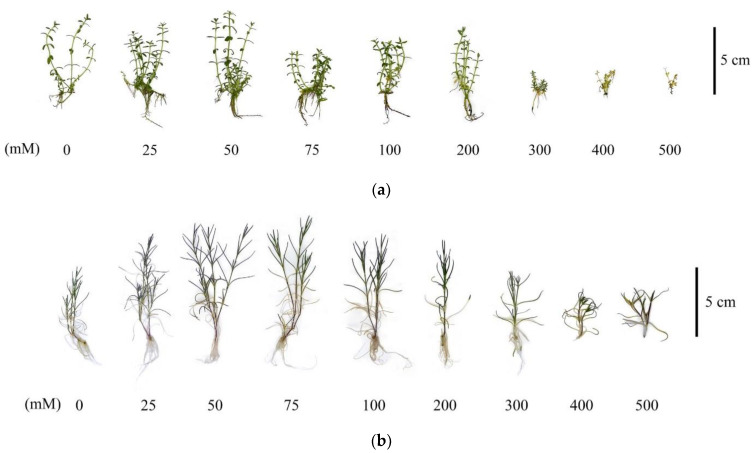
Characteristic morphological changes of *G. maritima* (**а**) and *S. marina* (**b**) explants after 40-day-long cultivation at different concentrations of NaCl in the medium.

**Figure 4 plants-12-01905-f004:**
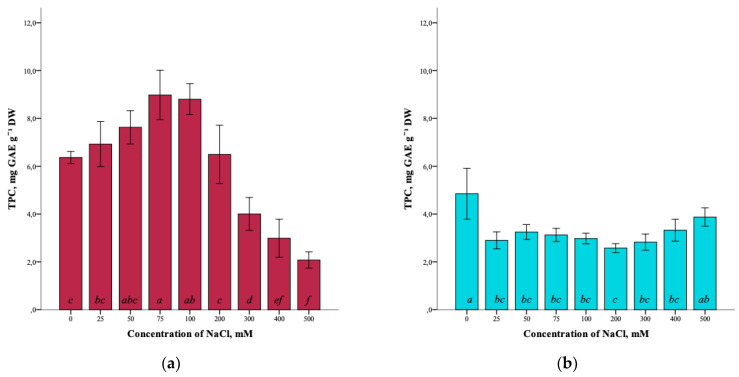
Effect of salinity stress on the total content of phenolic compounds (TPC) in *G. maritima* (**а**) and *S. marina* (**b**). Different letters indicate significant differences (ANOVA, Tukey-HSD, *p* ≤ 0.05, *n* = 4). DW—dry weight; GAE—gallic acid equivalents.

**Figure 5 plants-12-01905-f005:**
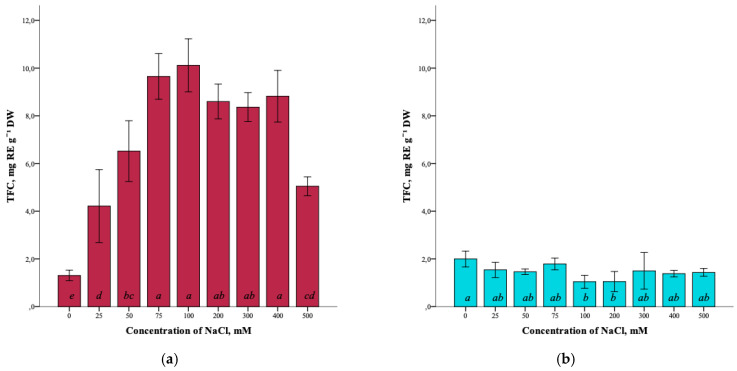
Effect of salinity stress on the total content of flavonoids (TFC) in *G. maritima* (**а**) and *S. marina* (**b**). Different letters indicate significant differences (ANOVA, Tukey-HSD, *p* ≤ 0.05, *n* = 4). DW—dry weight; RE—rutin equivalents.

**Figure 6 plants-12-01905-f006:**
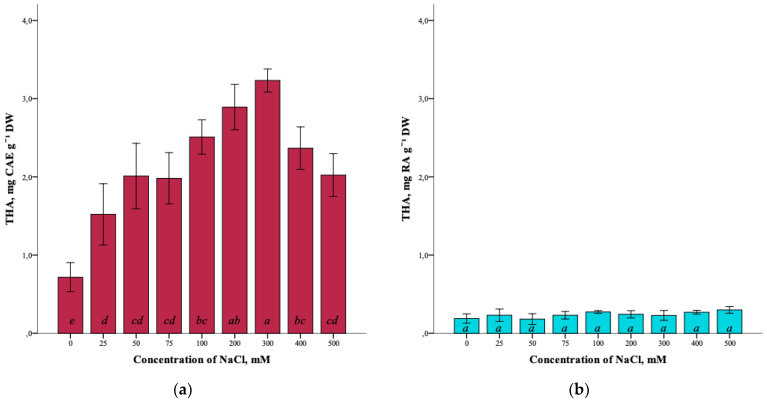
Effect of salinity stress on the total content of hydroxycinnamic acids (THA) in *G. maritima* (**а**) and *S. marina* (**b**). Different letters indicate significant differences (ANOVA, Tukey-HSD, *p* ≤ 0.05, *n* = 4). DW—dry weight; CAE—chlorogenic acid equivalents; RA—rosmarinic acid equivalents.

**Figure 7 plants-12-01905-f007:**
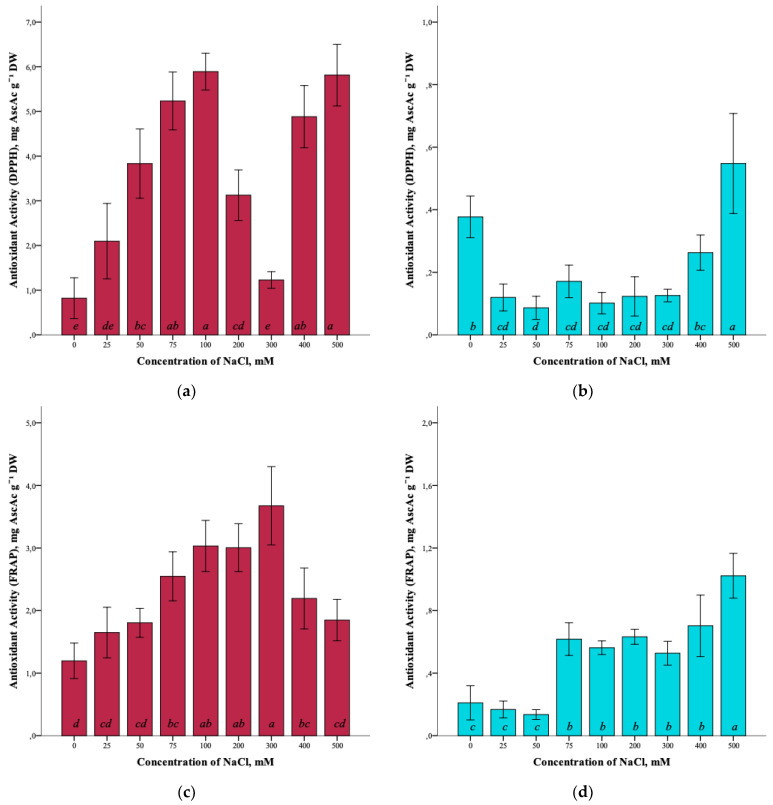
Effect of salinity stress on the antioxidant activity of extracts from *G. maritima* (**а**,**c**,**e**) and *S. marina* (**b**,**d**,**f**). Different letters indicate significant differences (ANOVA, Tukey-HSD, *p* ≤ 0.05, *n* = 4). DPPH—antioxidant activity determined by 2,2-diphenyl-1-picrylhydrazyl assay; ABTS—antioxidant activity determined by 2,2′-azino-bis(3-ethylbenzothiazoline-6-sulfonic acid) assay; FRAP—ferric reducing/antioxidant power; DW—dry weight; AscAc—ascorbic acid equivalent.

**Figure 8 plants-12-01905-f008:**
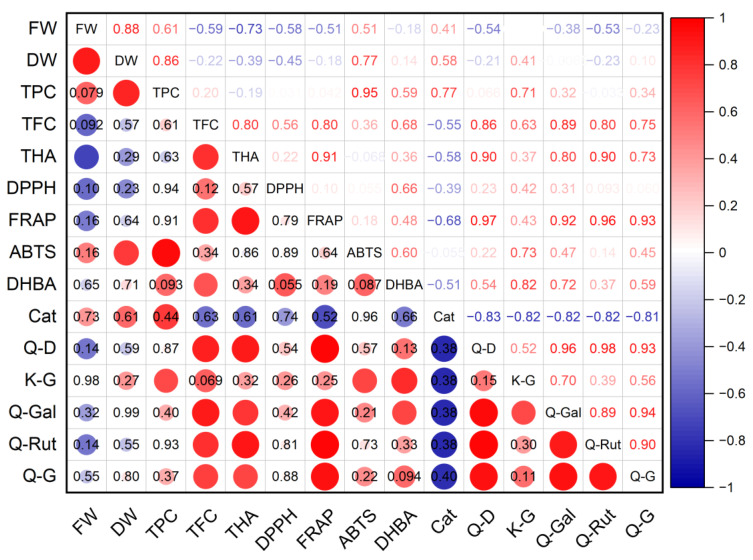
Correlation matrix with the Pearson coefficient values for phenolic compounds and antioxidant activity of *G. maritima*. In the upper-right part of the correlation matrix, Pearson’s correlation coefficients (−1 ≤ *r* ≤ 1) are presented; in the lower-left part, red or blue circles represent the significance of correlations—circles without labels indicate significant correlations (*p* ≤ 0.05); in other circles or cells, *p*-values of insignificant correlations (*p* > 0.05) are presented. ABTS—antioxidant activity determined by 2,2′-azino-bis(3-ethylbenzothiazoline-6-sulfonic acid) assay; Cat—catechin; DHBA—3,4-Dihydroxybenzoic acid (protocatechuic acid); DPPH—antioxidant activity determined by 2,2-diphenyl-1-picrylhydrazyl assay; DW—shoot dry weight; FRAP—ferric reducing/antioxidant power; FW—shoot fresh weight; K-G—kaempferol 3-*O*-glucoside (astragalin); Q-D—quercetin derivative; Q-G—quercetin 3-β-d-glucoside (isoquercitrin); Q-Gal—quercetin 3-galactoside (hyperoside); Q-Rut—quercetin 3-O-rutinoside (rutin); TFC—total flavonoid content; THA—total content of hydroxycinnamic acids; TPC—total phenolic content.

**Figure 9 plants-12-01905-f009:**
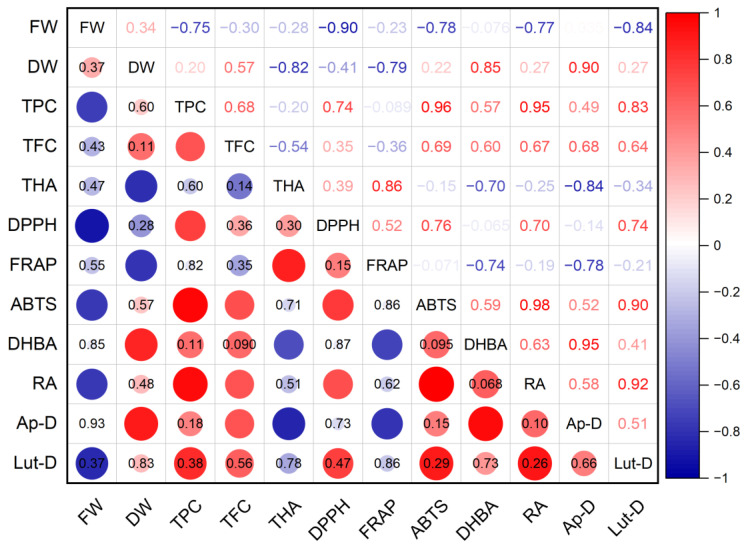
Correlation matrix with the Pearson coefficient values for phenolic compounds and the antioxidant activity of *S. marina*. In the upper-right part of correlation matrix, Pearson’s correlation coefficients (−1 ≤ *r* ≤ 1) are presented; in the lower-left part, red or blue circles represent the significance of correlations—circles without labels indicate significant correlations (*p* ≤ 0.05); in other circles or cells, *p*-values of insignificant correlations (*p* > 0.05) are presented. ABTS—antioxidant activity determined by 2,2′-azino-bis(3-ethylbenzothiazoline-6-sulfonic acid) assay; Ap-D—apigenin derivative; DHBA—3,4-Dihydroxybenzoic acid (protocatechuic acid); DPPH—antioxidant activity determined by 2,2-diphenyl-1-picrylhydrazyl assay; DW—shoot dry weight; FRAP—ferric reducing/antioxidant power; FW—shoot fresh weight; Lut-D—luteolin derivative; RA—rosmarinic acid; TFC—total flavonoid content; THA—total content of hydroxycinnamic acids; TPC—total phenolic content.

**Table 1 plants-12-01905-t001:** Content of individual phenolic acids and flavonoids in *G. maritima* depending on salinity. Statistical results are presented as mean ± standard deviation (*n* = 4).

Concentration of NaCl, mM	Content of Individual Phenolic Compounds, µg g^−1^ DW						
	3,4-Dihydroxybenzoic Acid (Protocatechuic Acid)	Catechin	Kaempferol 3-*O*-Glucoside (Astragalin)	Quercetin 3-Galactoside (Hyperoside)	Quercetin 3-*O*-Rutinoside (Rutin)	Quercetin 3-β-d-Glucoside (Isoquercitrin)	Quercetin Derivative *
0	10 ± 1 e **	148 ± 3 b	173 ± 7 d	44 ± 2 e	25 ± 2 f	10 ± 2 c	139 ± 3 f
25	17 ± 2 de	174 ± 8 a	187 ± 12 d	66 ± 5 e	46 ± 7 f	18 ± 3 c	251 ± 11 f
50	25 ± 3 c	–	942 ± 41 a	432 ± 25 c	252 ± 10 e	33 ± 3 c	2058 ± 78 e
75	34 ± 2 b	–	792 ± 24 b	812 ± 42 b	660 ± 31 c	159 ± 18 b	4454 ± 405 c
100	48 ± 4 a	–	974 ± 57 a	1062 ± 104 a	633 ± 51 cd	161 ± 15 b	5054 ± 343 b
200	25 ± 3 c	127 ± 2 c	723 ± 48 b	853 ± 52 b	865 ± 47 b	159 ± 10 b	5325 ± 294 b
300	19 ± 4 cde	–	395 ± 33 c	910 ± 37 b	1062 ± 117 a	192 ± 20 a	6256 ± 309 a
400	15 ± 2 ef	–	220 ± 17 d	460 ± 25 c	536 ± 26 d	33 ± 3 c	3074 ± 208 d
500	22 ± 2 cd	–	225 ± 26 d	205 ± 17 d	256 ± 29 e	12 ± 2 c	2094 ± 120 e

* The compound was identified by UV spectra and quantified by quercetin 3-glucoside. ** Different letters indicate significant differences between treatments for each compound (ANOVA with post hoc Tukey HSD Test, *p* ≤ 0.05). – These concentrations are said to be below the limit of detection (LOD).

**Table 2 plants-12-01905-t002:** Content of individual phenolic acids and flavonoids in *S. marina* depending on salinity. Statistical results are presented as mean ± standard deviation (*n* = 4).

Concentration of NaCl, mM	Content of Individual Phenolic Compounds, µg g^−1^ DW			
	3,4-Dihydroxybenzoic Acid (Protocatechuic Acid)	Rosmarinic Acid	Apigenin Derivative *	Luteolin Derivative
0	71 ± 2 a **	395 ± 9 a	2345 ± 131 a	281 ± 4 a
25	36 ± 2 b	158 ± 5 c	1649 ± 64 b	–
50	31 ± 2 c	144 ± 8 cd	1683 ± 65 b	–
75	29 ± 2 c	128 ± 8 d	1652 ± 69 b	–
100	30 ± 1 c	104 ± 7 e	1390 ± 103 c	–
200	19 ± 2 d	85 ± 9 ef	1366 ± 11 c	–
300	16 ± 1 d	73 ± 8 f	1437 ± 63 c	–
400	17 ± 1 d	99 ± 9 e	1101 ± 90 d	16 ± 1 c
500	3 ± 2 e	252 ± 17 b	984 ± 52 d	253 ± 13 b

* The compounds were identified by UV spectra and quantified by standard, with the same aglycon (apigenin 7-*O*-glucoside and luteolin 7-*O*-glucoside). ** Different letters indicate significant differences between treatments for each compound (ANOVA with post hoc Tukey HSD Test, *p* ≤ 0.05). – These concentrations are said to be below the limit of detection (LOD).

## Data Availability

The data presented in this study are available on request from the corresponding author.

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
