# Peer review of "Effect of Salinity Stress on Phenolic Compounds and Antioxidant Activity in Halophytes Spergularia marina (L.) Griseb. and Glaux maritima L. Cultured In Vitro"

_plants, 2023, doi:10.3390/plants12091905_

Round 1
Reviewer 1 Report
This paper is well organized in language and has obtained remarkable results. It is suggested to publish after minor revision.
Comments:
(1) It is better have a conclusion at the end of Abstract, such as in which range of NaCl concentrations, halophyte plants can obtain the most phenolic compounds. It will echoes the meaning of the first sentence in Abstract, and give a reference for further application.
(2) In the results, why did authors not calculate the content of phenolic compounds of per plant, in combination with dry biomass and concentration, which seems to make more sense.
(3) In the title of Figure 5, it is total of flavonoids, but not the total of phenolic compounds.
(4) add “of” between concentration and NaCl below the abscissa axis of Figures.
(5) Add the biological replicates (n) in Figure Legends.
(6) suggest to delete “of extracts” from the subtitle of 2.3.
Author Response
Response to Reviewer 1 Comments
We thank the Reviewer for the positive, thoughtful and straightforward comments.
All changes according to the notes of Reviewer have been highlighted in manuscript using the "Track Changes" function in Microsoft Word.
We address point by point the reviewer’s comments below:
Point 1: It is better have a conclusion at the end of Abstract, such as in which range of NaCl concentrations, halophyte plants can obtain the most phenolic compounds. It will echoes the meaning of the first sentence in Abstract, and give a reference for further application.
Response 1: The Abstract was corrected. Please see in text.
Point 2: In the results, why did authors not calculate the content of phenolic compounds of per plant, in combination with dry biomass and concentration, which seems to make more sense.
Response 2: Thank you very much for your valuable comment. We will take it into account in our future studies.
Point 3: In the title of Figure 5, it is total of flavonoids, but not the total of phenolic compounds.
Response 3: Thank you very much for your consideration. The error has been corrected.
Point 4: Add “of” between concentration and NaCl below the abscissa axis of Figures.
Response 4: All required Figures have been corrected.
Point 5: Add the biological replicates (n) in Figure Legends.
Response 5: Biological replicates (n) were added in the Legends to the Figures.
Point 6: Suggest to delete “of extracts” from the subtitle of 2.3.
Response 6: The text of the article has been corrected taking into account the comment.
Reviewer 2 Report
Dear authors,
I congratulate you for the work carried out, and I would ask you to take into account the following observations:
In the abstract, in lines 22, 23 and 36 before the numbers 75, 200 and 500 there is a hyphen that is interpreted as a negative value (minus 75, minus 200...), please remove this hyphen.
On line 126, what does GAE mean? What does DW mean?
In line 139 it incorporates the abbreviation RE and does not say what it means.
At the bottom of figure 5 you should say what the abbreviations TFC, RE mean.
In line 152 it incorporates the CAE abbreviation and does not say what it is going to be. Bottom of figure 6, what is THA, CAE...?
In the text before figure 7, it does not explain what the abbreviations DPPH, ABTS, FRAP mean. At the bottom of figure 7 if it is explained.
The first time you use an abbreviation you must specify what it means in order to read the text properly. In the captions of the figures, the meaning of the abbreviations must be included in order to be properly interpreted without resorting to the text where they are referenced.
You should correct Reference 2, it is misquoted.
All the best.
Author Response
Response to Reviewer 2 Comments
We thank the Reviewer for the positive, thoughtful and straightforward comments.
All changes according to the notes of Reviewer have been highlighted in manuscript using the "Track Changes" function in Microsoft Word.
We address point by point the reviewer’s comments below:
Point 1: In the abstract, in lines 22, 23 and 36 before the numbers 75, 200 and 500 there is a hyphen that is interpreted as a negative value (minus 75, minus 200...), please remove this hyphen.
Response 1: The text of the article has been corrected taking into account the comment.
Point 2: On line 126, what does GAE mean? What does DW mean?
Response 2: Abbreviation designations have been specified in the materials and methods in section 4.3.2. "Determination of the total content of certain groups of phenolic compounds". We also added a transcription of abbreviations in the text and in the legend of Figure 4.
Point 3: In line 139 it incorporates the abbreviation RE and does not say what it means. At the bottom of figure 5 you should say what the abbreviations TFC, RE mean.
Response 3: Abbreviation designations have been specified in the materials and methods in section 4.3.2. "Determination of the total content of certain groups of phenolic compounds". We also added a transcription of abbreviations in the text and in the legend of Figure 5.
Point 4: In line 152 it incorporates the CAE abbreviation and does not say what it is going to be. Bottom of figure 6, what is THA, CAE...?
Response 4: Abbreviation designations have been specified in the materials and methods in section 4.3.2. "Determination of the total content of certain groups of phenolic compounds". We also added a transcription of abbreviations in the text and in the legend of Figure 6.
Point 5: In the text before figure 7, it does not explain what the abbreviations DPPH, ABTS, FRAP mean. At the bottom of figure 7 if it is explained.
The first time you use an abbreviation you must specify what it means in order to read the text properly. In the captions of the figures, the meaning of the abbreviations must be included in order to be properly interpreted without resorting to the text where they are referenced.
Response 5: The text of the article has been corrected taking into account the comment.
Point 6: You should correct Reference 2, it is misquoted.
Response 6: Reference 2 has been corrected.
Reviewer 3 Report
In my opinion, the research is of interest, since it studies the antioxidant and metabolic response of two halophyte species (Spergularia marina and Glaux maritima) under controlled conditions at different salt concentrations. This work complements previous research already published in Plants with the same species and coinciding in many of the parameters analyzed. Although, it allows delving into the observed responses of both species in their natural habitats. The analytical description seemed quite correct to me, I have only indicated some small details. Although, at an experimental level I have not come to understand well what has been taken from the sample to carry out the statistical analysis, although the results seem to be expressed per plant. These aspects should be clarified to be more accessible to the reader. In addition, at a statistical level, I have observed that it should be more detailed in some aspects, and I have not come to understand the correlation matrix, far from being a classic correlation matrix, in which I believe the degree of significance is not indicated either. At the writing level, there are parts of the manuscript that are very good, but others, in my opinion, should be rewritten; that is, the entire manuscript must be reviewed. In particular, the introduction seemed to me to be improved, it was short and with little bibliography. In the same way, the discussion does not discuss in a linear way the results obtained with those found in the bibliography, nor with the previous works carried out. I think that the research should be published, despite the fact that it has not delved into determining what are the mechanisms of Glaux maritima that seem to be different from those of Spergularia maritima. I have indicated the main comments in the manuscript; although as I have already mentioned, the writing of the abstract, introduction, results and discussion should be improved. The English seemed fine to me. Conclusions must be included in the manuscript.

In my opinion, English is correct, only minor editing required. Although the expression can be improved in some parts of the text.
Author Response
Response to Reviewer 3 Comments
We thank the Reviewer for the positive, thoughtful and straightforward comments.
We corrected the Abstract, Introduction, Results, Discussion, Materials and Methods sections following the reviewer’s indications. Conclusions was also added to the manuscript.
All changes according to the notes of Reviewer have been highlighted in manuscript using the "Track Changes" function in Microsoft Word.
We address point by point the reviewer’s comments below:
Point 1: In my opinion, the research is of interest, since it studies the antioxidant and metabolic response of two halophyte species (Spergularia marina and Glaux maritima) under controlled conditions at different salt concentrations. This work complements previous research already published in Plants with the same species and coinciding in many of the parameters analyzed. Although, it allows delving into the observed responses of both species in their natural habitats.
The analytical description seemed quite correct to me, I have only indicated some small details.
Response 1: We took into account the comments and made corrections to the Introduction. Please see in text.
Point 2: Although, at an experimental level I have not come to understand well what has been taken from the sample to carry out the statistical analysis, although the results seem to be expressed per plant. These aspects should be clarified to be more accessible to the reader.
Response 2: We took into account the comments and made corrections to Materials and Methods (4.3.1. Extract preparation).
Point 3: In addition, at a statistical level, I have observed that it should be more detailed in some aspects, and I have not come to understand the correlation matrix, far from being a classic correlation matrix, in which I believe the degree of significance is not indicated either.
Response 3: The text of the article has been corrected taking into account the comment. We supplemented the legend of Figure 8 and Figure 9.
Point 4: At the writing level, there are parts of the manuscript that are very good, but others, in my opinion, should be rewritten; that is, the entire manuscript must be reviewed. In particular, the introduction seemed to me to be improved, it was short and with little bibliography.
Response 4: We took into account the comments and made corrections to the Introduction. The bibliography has been supplemented. Please see in text.
Point 5: In the same way, the discussion does not discuss in a linear way the results obtained with those found in the bibliography, nor with the previous works carried out.
Response 5: We took into account the comments and made corrections to Discussion.
Point 6: I think that the research should be published, despite the fact that it has not delved into determining what are the mechanisms of Glaux maritima that seem to be different from those of Spergularia maritima. I have indicated the main comments in the manuscript; although as I have already mentioned, the writing of the abstract, introduction, results and discussion should be improved. The English seemed fine to me. Conclusions must be included in the manuscript.
Response 6: Thank you very much for your valuable comments. Conclusions was also added to the manuscript.
Round 2
Reviewer 3 Report
Thank the researchers for the clarifications to my comments. In general, most of my suggestions have been included and my doubts have been answered. I think the article is sufficiently improved for publication. Although, I have included a small comment. However, I still think that the abstract is too long, and that the work should be strengthened bibliographically.

In my opinion, English is fine.
